# K103N, V106M and Y188L Significantly Reduce HIV-1 Subtype C Phenotypic Susceptibility to Doravirine

**DOI:** 10.3390/v16091493

**Published:** 2024-09-20

**Authors:** Nikita Reddy, Maria Papathanasopoulos, Kim Steegen, Adriaan Erasmus Basson

**Affiliations:** 1HIV Pathogenesis Research Unit, University of the Witwatersrand, Johannesburg 2193, South Africamaria.papathanasopoulos@wits.ac.za (M.P.); 2Department of Molecular Medicine and Haematology, School of Pathology, Faculty of Health Science, University of the Witwatersrand, Johannesburg 2193, South Africa; kim.steegen@nhls.ac.za

**Keywords:** HIV, NNRTI, doravirine, phenotypic, resistance, subtype C

## Abstract

Doravirine (DOR) is a non-nucleoside reverse transcriptase inhibitor (NNRTI) with efficacy against some NNRTI-resistant mutants. Although DOR resistance mutations are established for HIV-1 subtype B, it is less clear for non-B subtypes. This study investigated prevalent NNRTI resistance mutations on DOR susceptibility in HIV-1 subtype C. Prevalent drug resistance mutations were identified from a South African genotypic drug resistance testing database. Mutations, single or in combination, were introduced into replication-defective pseudoviruses and assessed for DOR susceptibility in vitro. The single V106M and Y188L mutations caused high-level resistance while others did not significantly impact DOR susceptibility. We observed an agreement between our in vitro and the Stanford HIVdb predicted susceptibilities. However, the F227L mutation was predicted to cause high-level DOR resistance but was susceptible in vitro. Combinations of mutations containing K103N, V106M or Y188L caused high-level resistance, in agreement with the predictions. These mutations are frequently observed in patients failing efavirenz- or nevirapine-based first-line regimens. However, they are also observed in those failing a protease inhibitor-based second-line regimen, as we have observed in our database. Genotypic drug resistance testing is therefore vital prior to the initiation of DOR-based treatment for those previously exposed to efavirenz or nevirapine.

## 1. Introduction

HIV and AIDS has been a public health issue since 1981 when the first cases were reported [1]. With 1.3 million new HIV infections and 630,000 AIDS-related deaths reported globally in 2023, HIV/AIDS is still a major concern [2]. The epidemic is particularly problematic in Africa, which accounts for more than two-thirds of the global population living with HIV [3]. South Africa has a high prevalence of HIV-1 (17.1%) with 7.5 million people living with HIV (PLH) in 2023 [4] and HIV-1 subtype C accounting for 98.44% of HIV infections [5,6,7]. Antiretroviral drugs (ARV) allow PLH to suppress viremia below detectable levels in the blood and provide PLH with an improved quality and span of life [8,9]. In addition, optimal antiretroviral therapy (ART) has been shown to reduce the transmission of HIV [10]. ART regimens generally comprise a combination of ARV drugs from two or more of the following classes: attachment/fusion inhibitors, nucleos(t)side reverse transcriptase inhibitors (NRTIs), non-nucleoside reverse transcriptase inhibitors (NNRTIs), integrase strand-transfer inhibitors (INSTIs) and protease inhibitors (PI). Consistent ART adherence may be difficult to achieve [11] and the sub-optimal use of ART could lead to the development of ARV drug resistance and treatment failure [12]. Resistance to NRTIs/NNRTIs, INSTIs and PIs is caused by mutations in reverse transcriptase, integrase and/or protease, respectively [13]. An increase in the prevalence of pre-treatment drug resistance (PDR) has been observed over the past decade globally [14]. High levels of PDR were also observed in South Africa during a national HIV household survey, where 15.3% of PLH who self-reported not to be on treatment and had undetectable ARV levels presented with drug resistance mutations (DRM) [15]. Only non-nucleoside reverse transcriptase inhibitor (NNRTI) resistance mutations were observed in these PL. The prevalence of NNRTI DRMs in patients who defaulted treatment was as high as 56.4%. In May 2019, the South African National ART Clinical Guidelines implemented Dolutegravir (DTG), an INSTI, for first- and second-line combination ART (cART). This circumvents the impact of NNRTI DRMs on treatment efficacy since these mutations do not impact DTG susceptibility. However, side effects [16] and weight gain [17,18] associated with DTG-based cART may lead PLH to seek alternative treatment options.

Doravirine (DOR) is a third-generated NNRTI which is effective against the three most prominent NNRTI DRMs: K103N, G190A and Y181C [19,20,21]. The major DOR-associated resistance mutations include V106AIM [22,23,24], Y188L [22,23,24] and F227L [25,26]. Other NNRTI DRMs that have also been observed to reduce DOR susceptibility include V108I, G190ES, H221Y, M230IL, L234I and P236L [20,22,25]. DOR is administered in a single daily oral dose [27], has an improved safety profile and has fewer drug interactions than other NNRTIs [28]. The DRIVE-FORWARD (NCT02275780) and DRIVE-AHEAD (NCT02403674) phase III clinical trials demonstrated a favourable DOR safety profile and fewer neuropsychiatric side effects and adverse events [29,30]. The development of DOR resistance in both trials, as well as their extensions, was low (≤2%) [31]. In the DRIVE-SWITCH (NCT02397096) trial, switching to a DOR-based regimen demonstrated maintained viral suppression for 2 years, a favourable safety profile, an improved lipid profile, minimal weight gains and the absence of drug resistance [32]. These clinical trials therefore support the use of DOR for long-term first-line cART and for virally suppressed PLH who wish to switch therapy. DOR is currently approved in Europe [33] and the United States [34] to treat ART naïve and virally suppressed PLH. While the drug has been approved in South Africa [35], it is currently not available in the public sector [36]. With the high levels of pre-treatment and acquired NNRTI drug resistance in South Africa, we aimed to investigate the magnitude of phenotypic drug resistance that prevalent NNRTI mutations, in the context of the HIV-1 subtype C, would pose to DOR. This will inform HIV clinicians and stakeholders in the public sector on the value of implementing DOR-based cART in South Africa and other parts of the world where the HIV-1 subtype C is prevalent. 

## 2. Materials and Methods

### 2.1. Vectors

The HIV-1 Gag-Pol expression vectors, p8.9NSX+ [37] and p8.9MJ4 [38] (both derivatives of pCMV∆R9 [39]), were obtained from Deenan Pillay (University College London, London, UK). These vectors contain HIV-1 *gag* and *pol*, which provide the internal structural components and enzymes for the resulting replication-defective pseudoviruses (PSVs). p8.9NSX+ encodes for HIV-1 subtype B *gag-pol*, in which protease and most of reverse transcriptase was replaced with that of the HIV-1 subtype C MJ4 (GenBank accession number AF321523.1) to yield p8.9MJ4. A *PvuI* endonuclease restriction site at nucleotide positions 10–15 (amino acid positions 4–6) and a *HpaI* endonuclease restriction site at nucleotide positions 1100–1106 (amino acid positions 366–369) (HXB2 nucleotides 3682–3688) were introduced in the reverse transcriptase of both vectors. Additionally, the *PvuI* endonuclease restriction site at the beginning of protease in p8.9NSX+ was silenced. The *PvuI* and *HpaI* restriction sites allowed for the introduction of a 1096 bp reverse transcriptase fragment from the laboratory-adapted strains into the vectors. The pHEF-VSVG (ARP-4693) envelope expression vector encodes for the Vesicular Stomatitis Virus protein G and was contributed by Dr. Lung-Ji Chang (NIH HIV Reagent Program, Division of AIDS, NIAID, NIH). The pCSFLW transfer vector was obtained from Dr. Nigel Temperton (University College London, London, UK) and contains the firefly luciferase reporter gene. The co-transfection of mammalian cells with all three vectors produced VSVG pseudotyped HIV-1-like virions that are replication defective and that introduce the firefly luciferase reported gene into the genome of infected cells. Viral infection and the inhibition of HIV-1 reverse transcriptase or integrase by ARVs can be monitored by quantifying the levels of luciferase expression in the cells.

### 2.2. Laboratory-Adapted Strains 

Wild-type HIV-1 subtype B laboratory-adapted strains (i.e., DS9, LTNP5, SM1, SM2), and subtype C laboratory-adapted strains (i.e., CM9, DU151, DU178, DU422) (Appendix A) were obtained from the National Institute for Communicable Diseases (NICD, Johannesburg, South Africa) as frozen culture supernatants. These strains were cloned into HIV-1 Gag-Pol expression vectors and selected NNRTI DRMs were introduced through SDM to assess DOR resistance.

### 2.3. HEK293T Cell Culture

The Human Embryonic Kidney 293T (HEK293T) continuous mammalian cell line was cultured in Dulbecco’s Modified Eagle’s Medium (DMEM) with 4.5 g/L glucose, L-glutamine and sodium pyruvate (PAN Biotech, Aidenbach, Germany, Cat. No. P04-03590). The medium was supplemented with 10% foetal bovine serum (Thermo Fisher Scientific, Waltham, MA, USA, Cat. No. 10493106), 40 mM HEPES (Thermo Fisher Scientific, Waltham, MA, USA, Cat. No. 15630056) and 50 mg/L gentamycin (Thermo Fisher Scientific; Cat. No. 15710049). The cells were maintained at 37 °C under 5% CO_2_ in a humidified incubator and passaged every 2–3 days.

### 2.4. Antiretroviral Drugs

DOR (MK-1439) was purchased as a powder from MedChemExpress (Middlesex, NJ, USA, CAS No. 1338225-97-0). The following reagent was obtained through the NIH HIV Reagent Program, NIAID, NIH: Efavirenz, HRP-4624, contributed by DAIDS/NIAID. Both DOR and Efavirenz (EFV) were prepared in dimethyl sulfoxide (DMSO) (Sigma-Aldrich, St. Louis, MO, CAS No. 67-68-5) and diluted in complete DMEM to their respective working concentrations.

### 2.5. Generation of Laboratory-Adapted Strain-Derived PSVs

Viral RNA was extracted from the laboratory-adapted strain cultures using the QIAmp Viral RNA Mini Kit (Qiagen, Hilden, Germany, Cat No. 52904) according to the manufacturer’s instructions. cDNA was prepared from the RNA using the SuperScript III First Strand Synthesis System (Thermo Fisher Scientific, Waltham, MA, USA, Cat. No. 18080-051) and primer 3660R (5′-CCATGGCTATTTTTTGCACTGC-3′) according to the manufacturer’s instructions. A portion of the reverse transcriptase (HXB2 nucleotides 2528–3681) was amplified using the Roche Expand™ High Fidelity PCR System (Roche, Basel, Switzerland, Cat No. 11732641001) and primers 2528F (5′-GCTTGGATGCACACTAAATTTTCC-3′) and 3660R on the C1000 Touch™ Thermal Cycler with 96-Well Fast Reaction Module (Bio-rad, Sandton, South Africa, Cat. No. 1851196). The first round of PCR reactions contained the following: 5.0 µL PCR buffer (10×), 1.0 μL primer 2528F (3 μM), 1.0 μL primer 3660R (3 μM), 1.0 μL dNTP mix (10 mM each), 0.8 μL polymerase (3.5 U/μL), 38.7 μL PCR-grade water and 2.5 μL cDNA reaction. The following thermal cycling conditions were used: 94 °C for 2 min; 10 cycles of 94 °C for 15 s, 55.5 °C for 30 s and 72 °C for 70 s; 20 cycles of 94 °C for 15 s, 55.5 °C for 30 s and 72 °C for 20 s (+5 min every cycle); and a final extension of 72 °C for 7 min. The *PvuI* and *HpaI* restriction sites were introduced into the 5′- and 3′-ends of the amplicons, respectively, during the second round of PCR using the TOPO^TM^ XL-2 PCR Cloning Kit (Thermo Fisher Scientific, Waltham, MA, USA, CAT No. 45-20810) with primers 2560F (5′-CGATCGAAACTGTACCAGTAAAATTAAAGC-3′) and 3628R (5′-GTTAACTGTTTTACATCATTAGTGTGGG-3′). The second round of PCR reactions contained the following: 25 μL of Platinum SuperFi Green PCR Master Mix (2×), 2.5 μL of primer 2560F (10 μM), 2.5 μL of primer 3628R (10 μM), 10 μL of SuperFi GC Enhancer (5×), 5.0 μL of PCR-grade water and 5.0 μL of first-round PCR reaction. The following thermal cycling conditions were used: 98 °C for 30 s, followed by 35 cycles of 98 °C for 10 s, 55.5 °C for 10 s and 72 °C for 1 min, with a final extension of 72 °C for 4 min. The PCR reactions were analysed on 1% agarose gel in Tris-Acetate-EDTA (TAE; ThermoFisher Scientific, Waltham, MA, USA, Cat. No. B49) buffer containing 0.5 ug/mL Ethidium Bromide Solution (Bio-rad, Sandton, South Africa, Cat. No. 1610433) and visualized under UV light for the amplicons of 1095 base pairs (bp) in size. The PCR products were cleaned up with the PureLink™ PCR Purification Kit (Invitrogen, Waltham, MA, USA, Cat. No. K310001) and the DNA was quantified using the NanoDrop 1000 Spectrophotometer (ThermoFisher Scientific; Cat. No. ND2000CLAPTOP). The purified amplicons were subjected to Sanger sequencing to confirm sequence identity and absence of cross-contamination during PCR. 

The amplicons were first cloned into the pCR-XL-2-TOPO vector (Thermo Fisher Scientific, Waltham, MA, USA, CAT No. 45-20810) according to the manufacturer’s instructions. DH5αbacterial cells were transformed with the reaction mixtures through standard heat shock and plated on carbenicillin-containing (100 µg/mL) LB agar plates. After incubation at 37 °C for approximately 16 h, colonies were picked and cultured overnight in carbenicillin-containing (100 µg/mL) LB broth. Bacteria were pelleted at 5000 rpm for 15 min in an Eppendorf 5810R Refrigerated Centrifuge (Eppendorf, Midrand, South Africa Cat. No. 5811000010). The supernatants were discarded, and plasmid DNA was extracted with the QIAprep Spin Miniprep Kit (QIAGEN, Hilden, Germany, Cat. No. 27104). The purified plasmids were subjected to Sanger sequencing to confirm the sequence identity of the inserts.

TOPO clones and HIV-1 Gag-Pol expression vectors (p8.9MJ4, p8.9NSX+) were double-digested with *PvuI*-HF (New England Biolabs (NEB), Ipswich, MA, USA, Cat No. R3150S) and *HpaI* (NEB, Ipswich, MA, USA, Cat. No. R0105S). The digested vectors were dephosphorylated using Shrimp Alkaline Phosphatase (NEB, Ipswich, MA, USA, Cat No. M0371S) to prevent re-circularization. The digested TOPO clones and vectors were electrophoresed on a 1% agarose gel in TAE-buffer containing ethidium bromide. Bands of the correct sizes were excised from the gel and purified with the Zymoclean Large Fragment DNA Recovery (Zymo Research, Orange, CA, USA, Cat. No. D4045). The excised TOPO inserts were phosphorylated with T4 Polynucleotide Kinase for the TOPO clones (NEB, Ipswich, MA, USA, Cat No. M0201S). The inserts were ligated into their respective HIV-1 Gag-Pol expression vector backbones (p8.9NSX+ for subtype B lad-adapted strains, p8.9MJ4 for subtype C laboratory-adapted strains) using the Quick Ligation Kit (NEB; Cat No. M2200) following the manufacturer’s instructions. DH5α bacterial transformation, LB-agar plating and culture, and plasmid DNA extraction followed. Sanger sequencing was used to confirm the sequence identity of the inserts.

### 2.6. Selection of NNRTI DRMs

HIV-1 reverse transcriptase sequences (*n* = 7749) from routine genotypic drug resistance testing, performed at the National Health Laboratory Services (NHLS, Charlotte Maxeke Johannesburg Academic Hospital, Johannesburg, South Africa) between January 2016 and December 2020, were screened for prevalent NNRTI DRMs. The NNRTI DRMs were defined as described by Wensing et al. (2022) [13]. As the South African National guidelines do not recommend genotypic drug resistance testing in patients failing first-line NNRTI-based cART [36], the sequences were obtained mostly from patients failing second-line protease inhibitor (PI)-based cART. The single and combinations of prevalent NNRTI DRMs were introduced into HIV-1 Gag-Pol expression vectors through site-directed mutagenesis (SDM) for in vitro phenotypic drug susceptibility testing.

### 2.7. Generation of NNRTI-Resistance Mutations

The selected single (*n* = 15) and combinations (*n* = 20) of NNRTI DRMs were generated in the original p8.9MJ4 vector through SDM. Most of the mutations were introduced using non-overlapping primers designed with NEBaseChanger version 1.3.3 (NEB, Ipswich, MA, USA,) (Appendix A). The PCR reactions were performed using the Q5 High-Fidelity 2×Master Mix (NEB, Ipswich, MA, USA, Cat. No. M0492S). The SDM reaction contained the following: 12.5 μL of master mix (2×), 1.3 μL of forward primer (10 μM), 1.3 of μL reverse primer (10 μM), 1.0 μL of vector DNA (25 ng/μL) and 8.9 μL of PCR-grade water. The reactions were carried out under the following conditions on the C1000 Touch™ Thermal Cycler with 96-Well Fast Reaction Module (Bio-rad, Sandton, South Africa, Cat. No. 1851196): 98 °C for 30 s, 25 cycles of 98 °C for 10 s, 50–72 °C (dependent on the Tm of the primers) for 30 s and 72 °C for 10 min and a final extension of 72° for 2 min. The PCR reactions were analysed on a 1% agarose gel and visualized under UV light to confirm the amplicons were of the correct size. The amplicons were phosphorylated with T4 Polynucleotide Kinase (ThermoFisher Scientific, Waltham, MA, USA, Cat No. EK0031, 10 U/µL), ligated to circularize with T4 DNA Ligase (ThermoFisher Scientific, Waltham, MA, USA, Cat No. EL0012, 5 U/µL), and the original template was digested using *DpnI* (ThermoFisher Scientific, Waltham, MA, USA, Cat No. ER1701, 10 U/µL). For problematic SDM reactions, overlapping primers were designed with QuickChange Primer Design (Agilent, Santa Clara, CA, USA) (Appendix A). The mutagenesis was performed according to Yang et al. (2022) [40]. PCR reactions were assembled as for the non-overlapping SDM PCR and the same thermocycling was used. DH5αbacterial cells were transformed with the ligation reaction, followed by plating on LB-agar and culture in LB-broth. The plasmid DNA was extracted with the QIAprep Spin Miniprep Kit. Sanger sequencing and the submission of the consensus sequences to the Standford HIVdb Program [41] were used to confirm the presence of the relevant mutation(s). For clones with more than one NNRTI DRM, single mutations were introduced in multiple rounds of SDM until all the mutations were present in the vector.

### 2.8. Production of HIV-1-like Pseudoviruses

The culture supernatant was removed from HEK293T cell culture flasks and the cells were washed with 5 mL of Dulbecco’s Phosphate-Buffered Saline (DPBS without calcium or magnesium; PAN Biotech, Aidenbach, Germany, Cat. No. P04-36500). The cells were dislodged from the culture flask surface by incubating for 10 min at room temperature with 1 mL TrypLE™ Express Enzyme (ThermoFisher Scientific, Waltham, MA, USA, Cat. No. 12604021) and then resuspended in 5 mL complete DMEM. Viable cells were enumerated using the Countess™ II Automated Cell Counter (ThermoFisher Scientific, Waltham, MA, USA, Cat No. AMQAX1000). Transfection dishes (Nunclon™ Delta, 100 × 15 mm; ThermoFisher Scientific, Waltham, MA, USA, Cat. No. 150350) were prepared by plating with 8 × 10^6^ viable HEK293T cells in 10 mL of complete DMEM and incubating overnight at 37 °C under 5% CO_2_ in a humidified incubator. Transfection mixtures were prepared by combining the relevant wild-type or mutant HIV-1 Gag-Pol expression vector (1 µg) with pHEF-VSVG (0.25 µg) and pCSFLW (1.5 µg) in 100 µL of Opti-MEM™ Reduced-Serum Medium (Thermo Fisher Scientific; Cat. No. 31985062). After the addition of 12 µg of Polyethylenimine “Max” (PEI MAX, Polysciences, Warrington, PA, Cat. No. 24765) and incubation for 20 min at room temperature, the transfection mixture was added dropwise to the transfection dishes. After incubation for 48 h at 37 °C under 5% CO_2_ in a humidified incubator, the PSV-containing culture supernatants were harvested, filtered through a 0.45 µm syringe filter, aliquoted and stored at −80 °C. PSVs were titrated in 2-fold serial dilutions in complete DMEM (50 µL) on a Nunc™ Edge™ 96-Well, Nunclon Delta-Treated, Flat-Bottom Microplates (ThermoFisher, Waltham, MA, USA, Cat. No. 167425). After the addition of HEK293T cells (2 × 10^4^ cells/50 µL), the microplate was incubated at 37 °C under 5% CO_2_ in a humidified incubator for 48 h. The expression of firefly luciferase in the HEK293T cells was quantified with the Bright-Glo™ Luciferase Assay System (Promega, Madison, WI, Cat. No. E2620) on the GloMax^®^ 96 Microplate Luminometer (Promega, Madison, WI, Cat. No. E6501). PSV dilutions that produced a relative light unit (RLU) readout of 1 × 10^6^ were used in subsequent in vitro phenotypic assays.

### 2.9. In Vitro Phenotypic DOR Susceptibility Testing

For the in vitro phenotypic assays, eleven three-fold serial dilutions of DOR were prepared in duplicate in complete DMEM (50 µL) in Nunc™ Edge™ 96-Well Plates, staring at 1 µM DOR. Two wells contained complete DMEM only and served as the no-drug control. HEK293T cells were prepared at 4 × 10^5^ cells/mL and the PSV was added to the cells at the standardized titre. Fifty microliters of the mixture were then added to the wells of the plate. The final DOR concentrations ranged from 0.51 µM to 8.5 × 10^−6^ µM. EFV was included as inhibition control (0.2 µM to 3.4 × 10^−6^ µM) for the wild-type PSV. The plates were incubated at 37 °C under 5% CO_2_ in a humidified incubator for 48 h. The expression of firefly luciferase in the HEK293T cells was then quantified with the Bright-Glo™ Luciferase Assay System (Promega; Cat. No. E2620) on the GloMax^®^ 96 Microplate Luminometer (Promega; Cat. No. E6501). Each PSV was screened in at least two independent assays. Viral activity was calculated by dividing the RLU for each DOR concentration by the RLU of the no-drug control. A dose–response curve was constructed in Microsoft Excel (Microsoft, Redmond, Washington, DC, USA). The concentration of DOR that reduced the viral activity by 50% (IC_50_) was calculated using the ‘FORECAST’ formula in Microsoft Excel. The level of DOR resistance was expressed as the fold change (FC) in IC_50_ of the mutant PSV relative to the IC_50_ of the wild-type PSV. 

The lower technical cut-off (TCO) was defined as the 99th percentile of the average IC_50_ of the wild-type PSV determined over multiple phenotypic assays. DOR susceptibility/resistance was classified as follows: FC ≤ TCO = susceptible; TCO > FC ≤ 2 × TCO = potential low-level resistance; 2 × TCO > FC ≤ 3 × TCO = low-level resistance; 3 × TCO > FC ≤ 4 × TCO = intermediate resistance; and FC > 4 × TCO = high-level resistance. These cut-offs were not indicative of clinical outcomes but merely served as a way to classify the levels of DOR resistance in our assay. Assay-defined classifications were compared to the DOR drug resistance scores on Stanford’s HIV Drug Resistance Database [41,42,43]. Mutations with major or impactful discrepancies between the assay-defined classifications and Stanford’s drug resistance interpretation were introduced into the HIV-1 subtype B Gag-Pol expression vector (i.e., p8.9NSX+). The resulting PSVs were tested in vitro and the IC_50_ and FC values were compared to that of the related subtype C mutant.

### 2.10. Statistics

The average and standard deviation of the resulting FC and IC_50_ values from the DOR in vitro phenotypic testing of the replicates of each PSV were calculated using Microsoft Excel (Microsoft, Redmond, Washington, DC, USA). Intra-subtype variation was assessed using the FC values in a parametric unpaired Student’s *t*-test with Welch’s correction, which does not assume equal standard deviation. Inter-subtype variation was examined using the IC_50_ values and a Brown–Forsythe and Welch one-way ANOVA. Statistical significance was defined as a resulting *p*-value < 0.05 from statistical analysis. Statistical tests were performed using GraphPad version 8.0.2.263 (Dotmatics, Boston, MA, USA).

## 3. Results

### 3.1. Selection of NNRTI DRMs

Among the 7749 sequences, 982 sequences were from patients failing NNRTI-based regimens (12.7%). Most sequences were obtained from patients failing PI-based regimens (*n* = 5493; 70.9%). A proportion of sequences (*n* = 1152; 14.9%) were obtained from patients failing unknown regimens since the treatment regimen was not specified on the resistance test request forms. The remainder of the sequences were obtained from patients failing integrase strand-transfer inhibitor (INSTI)-based (*n* = 65; 0.8%) regimens or regimens where the complete ARV regimen was not indicated (*n* = 57; 0.7%).

NNRTI-associated drug resistance mutations were observed in 76.5% (*n* = 5928) of sequences. The most prevalent (*n* = 20) NNRTI DRM combinations were identified. These were constituted by various combinations of 14 different NNRTI DRMs. When detected as single NNRTI DRMs, their prevalence ranged from 14.6% (K103N) to 0.02% (L100I). The K103N mutation was observed in 60% (*n* = 12/20) of the NNRTI DRM combinations, with the most prevalent combination being K103N+P225H (6.5%). Although not frequently observed as a single mutation (0.4%), the V106M mutation was prevalent in combination with other NNRTI DRMs (*n* = 9/20, 45%). The Y181C NNRTI DRM is well-documented to cause intermediate to high-level resistance to other NNRTIs (EFV [44], NVP [44], RPV [45], ETR [46]), particularly in combination with other NNRTI DRMs. In addition, the mutation has a high prevalence in patients failing first-line NNRTI-based therapy in South Africa [47,48,49]. Although this mutation was not observed among the NNRT DRM combinations in sequences that we analysed, we did observe the single Y181C mutation at a low frequency (0.9%). The K101P and F227L mutations were not observed in isolation but only in combination with other NNRTI DRMs. All three of the latter mutations (K101P, Y181C, F227L) were included amongst the single NNRTI DRMs that were introduced into p8.9MJ4 for in vitro phenotypic testing.

### 3.2. In Vitro Phenotypic DOR Susceptibility Testing

#### 3.2.1. DOR Technical Cut-Off and Assay-Defined Classifications

The DOR technical cut-off values for wild-type subtype B (i.e., NSX) and subtype C (i.e., MJ4) PSVs were calculated using the IC_50_ values obtained over 20 and 30 independent assays, respectively. The average IC_50_ values for wild-type subtype B and subtype C PSVs were 0.0033 µM (±0.0011 SEM) and 0.0056 µM (±0.0022 SEM), respectively (Figure 1). DOR was more potent in wild-type NSX than MJ4, as indicated by MJ4’s significantly higher IC_50_ value (*p*-value < 0.0001). A significant difference, however, was not observed when comparing their FC values. The upper technical cut-off values were placed at the 99th percentile of the respective IC_50_ values: subtype B (0.0051 µM, 1.55 FC), and subtype C (0.0073 µM, 1.31 FC). The FC assay-defined classifications for subtype B were as follows: susceptible ≤ 1.55; potential low-level resistance > 1.55 to 3.09; low-level resistance > 3.09 to 4.64; intermediate resistance > 4.64 to 6.18; and high-level resistance > 6.18. The FC assay-defined classifications for subtype C were as follows: susceptible ≤ 1.31 FC; potential low-level resistance > 1.31 to 2.62 FC; low-level resistance > 2.62 to 3.93 FC; intermediate resistance > 3.93 to 5.24 FC; and high-level resistance > 5.24 FC.

#### 3.2.2. DOR Susceptibility of Single NNRTI DRMs

Full DOR susceptibility was observed in four PSV with single NNRTI DRMs: K101P (0.85 ± 0.20 FC), K103N (0.96 ± 0.02 FC), V179D (0.64 ± 0.03 FC) and Y181C (0.35 ± 0.07 FC) (Figure 2, Appendix A). Statistically significant decreases in FC relative to the wild-type were observed for V179D (*p* < 0.0001) and Y181C (*p*-value = 0.014), indicating increased susceptibility to DOR. L100I (1.36 ± 0.5 FC), K101E (1.79 ± 0.51 FC), K103S (1.23 ± 0.29 FC), E138A (1.56 ± 0.16 FC) and F227L (1.34 ± 0.24 FC) ranged from susceptibility to potential low-level resistance to DOR, with only F227L showing a statistically significant increase in FC (*p*-value = 0.031). A98G (2.99 ± 0.02 FC), V108I (2.84 ± 0.21 FC) and G190A (2.52 ± 0.55 FC) showed low-level DOR resistance, with the difference in FC being the most significant for A98G (*p*-value < 0.0001) compared to V108I (*p*-value = 0.046) and G190A (*p*-value = 0.041). Potential low to high-level DOR resistance was observed in P225H (3.86 ± 1.60 average FC) but it was not statistically different from the wild-type due to variability in the IC_50_ values. Both V106M (17.33 ± 5.31 FC) and Y188L (89.16 ± 0 FC) showed high-level resistance to DOR, and FC values were significantly higher than that of the wild-type (*p*-value = 0.001 and <0.0001, respectively) and other mutations (*p*-value = 0.0005 and <0.0001).

The phenotypic responses, based on the assay-defined classifications, were compared to the DRM scores calculated by the Stanford HIV Drug Resistance Database. The calculated DRM scores are shown in Figure 2. The in vitro phenotypic results were generally in agreement with the Stanford HIV Drug Resistance Database’s predictions; however, there were several notable contradictions. V106M displayed high-level DOR phenotypic resistance (17.33 ± 5.31 FC) in vitro, while Stanford predicted intermediate resistance. Y181C was phenotypically (hyper-)susceptible to DOR (0.35 ± 0.07 FC) despite predicted low-level resistance by Stanford HIVdb. A striking difference between the in vitro and predicted response was observed for F227L, which displayed low-level DOR resistance (1.34 ± 0.24 FC) in vitro compared to Stanford HIVdb, which predicted high-level resistance. To investigate whether the discrepancies for V106M and F227L could be strain- or subtype-related, these single mutations were introduced into HIV-1 subtype B and subtype C laboratory-adapted PSVs and screened in vitro. 

For the V106M single mutation, all subtype B strains containing V106M displayed statistically significant higher FC values compared to the wild-type: DS9 = 13.40 ± 1.66 FC; LTNP5 = 30.69 ± 2.75 FC; SM1 = 20.55 ± 4.31 FC; SM2 = 26.93 ± 5.63 FC; and NSX = 12.69 ± 1.86 FC (Figure 3, Appendix A). All subtype C strains containing V106M also displayed statistically significant higher FC values compared to the wild-type: CM9 = 23.59 ± 2.36 FC; DU151 = 7.73 ± 0.64 FC; DU179 = 37.67 ± 9.02 FC; DU422 = 29.46 ± 4.03 FC; and MJ4 = 17.33 ± 5.31 FC (Figure 3, Appendix A). To observe any strain-related differences in V106M resistance to DOR, the IC_50_ values of the laboratory-adapted strains with V106M were compared in a one-way ANOVA (Figure 4). The mutant CM9 PSV (0.132 ± 0.013 µM) was significantly less susceptible to DOR than the mutant DS9 PSV (0.044 ± 0.005 µM; *p*-value = 0.022) and mutant NSX PSV (0.042 ± 0.006 µM; *p*-value = 0.019) PSVs. The mutant DU151 PSV (0.043 ± 0.004 µM) was significantly less susceptible to DOR than the mutant LTNP5 PSV (0.101 ± 0.009 µM; *p*-value = 0.026). 

For the F227L single mutation, most subtype B strains containing F227L displayed statistically significant higher FC values compared to the wild-type: LNTP5 = 3.75 ± 1.01 FC; SM1 = 2.45 ± 0.17 FC; and SM2 = 3.90 ± 0.35 FC (Figure 5, Appendix A). Mutant DS9 PSV (1.30 ± 0.11) and mutant NSX PSV (0.97 ± 0.17) were both susceptible. All subtype C strains with F227L were susceptible: CM9 = 1.35 ± 0.11 FC; DU151 = 0.91 ± 0.11 FC; DU179 = 0.26 ± 0.07 FC; and DU422 = 0.69 ± 0.14 (Figure 5, Appendix A). The mutant DU179 PSV showed a statistically significant (*p*-value < 0.0001) increased susceptibility to DOR compared to the subtype C wild-type reference PSV (i.e., MJ4). Regarding strain-related differences in F227L resistance to DOR, both mutant SM1 PSV (0.008 ± 0.001 µM) and mutant SM2 PSV (0.013 ± 0.001 µM) had higher IC_50_ values compared to most of the mutant subtype C PSVs (0.001–0.008 µM; *p*-value 0.048–0.001) (Figure 6, Appendix A). The mutant DS9 PSV (0.004 ± 0.0003 µM) had a significantly lower IC_50_ than the mutant CM9 PSV (0.008 ± 0.001 µM; *p*-value = 0.035) but a significantly higher IC_50_ than the mutant DU179 PSV (0.001 ± 0.000 µM; *p*-value = 0.005). The mutant CM9 PSV had a significantly higher IC_50_ than the mutant NSX PSV (0.003 ± 0.001; *p*-value = 0.008). The mutant NSX PSV had a significantly lower IC_50_ than the mutant MJ4 PSV (0.008 ± 0.001; *p*-value = 0.015).

#### 3.2.3. DOR Susceptibility of Combined NNRTI DRMs 

High-level resistance to DOR was observed in 85% (*n* = 17/20) of the PSVs with combinations of NNRTI DRMs (Figure 7, Appendix A). These PSVs mostly contained combinations with K103N (*n* = 7/20, 35%), V106M (*n* = 6/20, 30%) or contained K103N + V106M (*n* = 2/20, 10%). The FC of PSVs containing the following combinations with K103N were significantly higher than the wild-type PSV and caused high-level resistance to DOR: A98G + K103N = 8.1 ± 2.3 FC; A98G + K103N + P225H = 80.5 ± 14.8 FC; L100I + K103N = 6.3 ± 1.5 FC; L1001 + K103N + P225H = 29.3 ± 10.2 FC; K103N + V108I+P225H = 17.5 ± 0.3 FC; K103N + Y188L > 89.2 FC; and K103N + P225H = 9.7 ± 2.2 FC. The K103N + V108I combination caused a low level of resistance to DOR (2.8 ± 0.3 FC), while K101P + K103N (0.8 ± 0.1 FC) and K103N + E138A (1.2 ± 0.2 FC) were susceptible. The following PSVs, containing combinations with V106M, caused high-level resistance to DOR with FC values greater than the upper limit of the assay (>89.2 FC): K101E + V106M + G190A; K103S + V106M; V106M + V179D + F227L; V106M + Y188L; V106M + G190A + F227L; and V179D + Y188L. The PSV that contained V106M + V179D also caused high-level DOR resistance (6.7 ± 0.3 FC). The two PSVs that contained combinations with K103N+V106M had FC values significantly higher than the wild-type PSV and caused high-level resistance to DOR: K103N + V106M = 81.9 ± 12.5 FC and K103N + V106M + F227L > 89.2 FC. All combinations with V106M were highly resistant to DOR. Interestingly, P225H and F227L were frequently observed in combination with K103N or V106M and caused high-level DOR resistance. The P225H mutation was always associated with K103N while the F227L mutation was always associated with V106M. Apart from combinations with K103N or/and V106M, the V179D + Y188L combination also caused high-level resistance to DOR, with FC values greater than the upper limit of the assay.

The phenotypic responses of the PSVs with NNRTI DRM combinations were compared to the DRM scores calculated by the Stanford HIV Drug Resistance Database (Figure 7). The database calculated high drug resistance scores for most of the combinations and agreed with what was observed in vitro. However, for 10 combinations, the database predicted intermediate DOR resistance while our assay-defined classifications awarded them with high-level resistance: A98G + K103N + P225H; L100I + K103N + P225H; K103N + V106M; K103N + V106M + F227L; K103N + V108I + P225H; K103N + Y188L + P225H; K101E + V106M + G190A; K103S + V106M; V106M + V179D; and V106M + G190A. In addition, the database predicted low-level resistance for the PSVs containing A98G + K103N and K103N + V108I, while these combinations showed high-level resistance and susceptible phenotypes, respectively. For K108N+V108I and A98G + K103N, both the database and the in vitro data were in agreement regarding a low-level resistance and susceptible phenotype, respectively.

## 4. Discussion

To ascertain the value of DOR-based treatment in the context of the NNRTI-experienced HIV-1 subtype C population in South Africa, this study performed in vitro phenotypic DOR susceptibility testing of HIV-1-like PSVs containing one or more NNRTI DRMs. Prevalent NNRTI DRMs were identified from a cohort of sequences obtained mostly from patients failing a second-line PI-based regimen. 

This study used a genotypic drug resistance database obtained from PLH who mostly failed second-line PI-based therapy between January 2016 to December 2020. During this period, PLH were initiated on NNRTI (EFV/NVP)-based ART and switched to boosted PI (LPV/r or ATV/r)-based second-line ART upon first-line failure [50]. The guidelines that recommended the initiation of PLH on an INSTI (DTG)-based first-line regimen were only published in May 2019 [51]. Therefore, the majority of PLH failing second-line ART during the study period would have been exposed to EFV/NVP before being switched to the PI-based regimen. As mentioned before, the South African National HIV Treatment Guidelines do not recommend genotypic drug resistance testing after failing on an NNRTI-based regimen. As such, the majority of sequences (70.9%) were obtained from PLH failing PI-based regimens. However, it is well-documented that NNRTI DRM can persist for extended periods of time after the cessation of NNRTI exposure [52,53,54]. This is most likely because NNRTI DRMs typically have minimal impact on viral replication fitness [55]. When referring to studies that have performed genotypic drug resistance testing on PLH that failed NNRTI-based ART in South Africa [48,49], we observed similar NNRTI DRM mutations were present in our sequence cohort. We did not observe V90I, Y188C, K101HP, V106A/I, Y181C, Y188C, G190S, H221Y, M230L or N348I in isolation or in combination with other NNRTI DRMs in our sequence cohort. However, these NNRTI DRMs were generally present at a prevalence < 12% in the referenced studies and this could explain why they were not observed at a later time point after NNRTI cessation in the PLH in our study. Of these NNRTI DRMs, M230L is the only mutation implicated in DOR resistance but is observed at low prevalence (0.2–4.4%) in PLH with HIV-1 subtype C failing NNRTI-based ART [49,56,57]. Thus, although the sequences in the current study were not obtained from PLH directly failing NNRTI-based cART, we do consider the NNRTI mutations that were screened in this study to be representative of those PLH that do directly fail NNRTI-based cART.

PSVs containing the single V106M or single Y188L mutations showed significantly decreased susceptibility to DOR, while the susceptibility of the single P225H-containing PSV varied from potential low to high-level DOR resistance. Due to discrepancies between the observed and predicted responses to DOR, the V106M and F227L single mutations were introduced into HIV-1 subtype B and subtype C laboratory-adapted PSVs and screened for DOR susceptibility. The V106M NNRTI DRM showed high-level resistance to DOR in all PSVs, confirming our initial screen. Contrary to the prediction, the F227L-containing HIV-1 subtype C PSVs remained susceptible to DOR while HIV-1 subtype B PSVs showed susceptible to low-level resistance phenotypes to DOR. Strain-specific differences were also observed for both V106M and F227L. PSVs with combinations of NNRTI DRMs containing K103N, and particularly V106M, showed high-level resistance to DOR. The P225H and F227L NNRTI DRMs were also frequently observed within combinations with K103N or V106M, and an association between P225H with K103N and F227L with V106M seemed evident. No major discrepancies were observed between the predicted and in vitro responses for NNRTI DRM combinations, except that higher levels of DOR resistance were observed in vitro.

The majority of the single NNRTI mutations displayed susceptibility or low-level resistance to DOR, including the highly prevalent K103N, Y181C and G190A NNRTI DRMs. However, high-level resistance was observed for the single V106M and Y188L mutations. Both were also predicted to cause reduced susceptibility to DOR by Stanford and are indicated as major DOR DRMs by the International AIDS Society USA [13]. V106A/M is selected for by EFV and NVP [44], with alanine (A) selected for at this position in subtype B, while methionine (M) is selected by subtype C [58,59]. V106M has been noted as a major DOR mutation in previous studies and is preferentially selected for by DOR [22,23,24]. Y188L is selected by NVP and EFV [60,61] and is also a major mutation selected by DOR [22,23,24]. We have therefore confirmed that these three single NNRTI DRMs also cause resistance to DOR in HIV-1 subtype C. Since the single V106M and Y188L mutations were highly resistant to DOR, it seems obvious that combinations with these NNRTI DRMs would also be highly resistant to DOR [25,26], as we have confirmed in our study.

The V179D and Y181C NNRTI SDMs displayed increased (hyper)susceptibility to DOR. Asante-Appiah et al. (2021) noted that clinical isolates (assumed to be HIV-1 subtype B) containing V179D had the lowest FC (0.5, 0.4–0.6 IQR) amongst 24 single NNRTI DRMs [62]. The Y181C NNRTI SDM had a susceptible phenotype (1.6 FC, 1.2–1.8 IQR) but did not show increased (hyper)susceptibility to DOR in the referenced study. Our observation regarding Y181C may be subtype-specific and will be explored further in future studies with clinical specimens that contain the mutation.

We observed a discrepancy between our in vitro data and the predicted response for the single F227L NNRTI DRM. Testing the mutation in HIV-1 subtype B and subtype C isolates showed that mutant subtype C strains tended to be more susceptible to DOR than mutant subtype B strains. We also observed significant strain-specific differences in DOR FC within both subtypes for this mutant, but neither subtype’s in vitro responses agreed with the predicted high-level resistance. In fact, significant differences in DOR IC_50_ values were also observed between the wild-type NSX and MJ4 PSVs, suggesting that the slight differences observed between the subtypes may be negligible for this mutation. Additionally, while differences were observed between mutant subtypes B and C laboratory-adapted strains, a direct comparison of clinical isolates may yield different phenotypes. Despite this, the difference between in vitro response of both subtypes with the contradictory high-level resistance predicted response by the Stanford HIV Drug Resistance Database is clear. However, the detailed RT phenotype query for DOR in the Stanford Database (Version 3.5.0 21 March 2024; 1 November 2023) only lists isolates with F227L in combination with other NNRTI DRMs (mostly V106A) and none with F227L alone. All the isolates had FC values ≥ 100 for DOR. We have also shown in this study that combinations with F227L (containing V106M) caused high-level resistance to DOR. The F227L and V106A NNRTI DRMs are often co-selected on NVP-based regimens [61]. Thus, the predicted response of the single F227L mutation to DOR is most likely exaggerated due to the presence of other NNRTI DRM in the isolates on which the drug resistance scores are based. 

The single P225H mutation showed a range of FC values (2.26–5.46 FC) amongst repeat in vitro screens, indicating low- to high-level resistance to DOR based on our assay-defined classifications. Because this variation was higher than the assay variation for the wild-type reference isolate (i.e., 1.31 FC), it may indicate that the variation for P225H is not assay-related. It is unclear whether this observation may be related to subtype, isolate or alternative binding mechanisms of DOR with the mutant HIV-1 reverse transcriptase binding pocket. However, based on the average FC (3.86), our classification was in agreement with Stanford’s classification (i.e., low-level DOR resistance). As we also observed in our study that the P225H mutation is frequently selected for alongside the prevalent K103N NNRTI DRM [60,61,63] and causes high-level resistance to DOR [64]. Future studies using clinical HIV-1 subtype C isolates containing the single (and combinations with) P225H NNRTI DRM will clarify the impact of this mutation on DOR susceptibility.

The single V106M and Y188L NNRTI DRMs, as well as combinations with K103N and V106M, were prevalent in our cohort and caused high-level resistance to DOR in vitro. 

Due to pre-treatment and acquired NNRTI drug resistance in the pre-DTG era, the initiation of, or switching to, a DOR-based ART in South Africa would not currently be effective for PLH. Genotypic drug resistance testing would likely be required to identify NNRTI-resistant mutations prior to initiation or switching to DOR. Since genotypic drug resistance testing is only recommended before switching to a third-line regimen in South Africa, DOR would most likely be more suitable as an option in third-line ART. However, the presence of archived NNRTI DRMs that could impact on DOR susceptibility in this context should be considered. The in vitro phenotypic testing of clinical isolates with linked clinical outcomes on DOR-based treatment is needed to assist HIV clinicians and drug resistance interpretation algorithms to accurately assess DOR resistance in clinical settings.

## Figures and Tables

**Figure 1 viruses-16-01493-f001:**
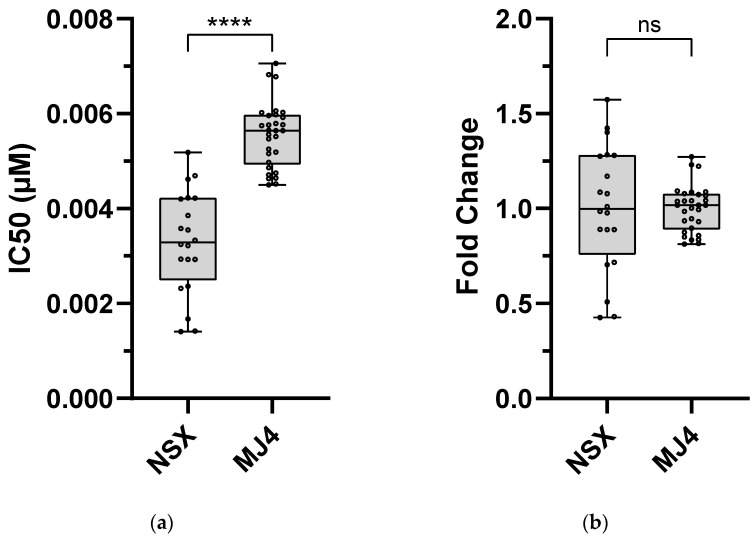
Phenotypic susceptibility of HIV-1 wild-type subtype B (NSX) and C (MJ4) to DOR. The HIV-1 subtype B (p8.9NSX+) and C (p8.9MJ4) backbone plasmids were used to produce the respective PSVs. In vitro phenotypic assays were performed to assess the susceptibility of each wild-type ubtype. (**a**) The average of the resulting IC_50_ values of the subtype B and C PSVs were used to calculate the FC for each mutant PSV. (**b**) DOR susceptibility was expressed as the IC_50_ value of the PSV compared to the average IC_50_ value of the corresponding subtype wild-type reference. The TCO was the average wild-type IC_50_ of the PSV compared to the 99th percentile of the wild-type IC_50_. The average IC_50_ values were as follows: subtype B—0.0033 µM; subtype C—0.0056 µM. The 99th percentile of the IC_50_ values were as follows: subtype B—0.0051 µM; subtype C—0.0073 µM. The TCOs were as follows: subtype B—1.55; subtype C—1.31. The TCOs were used to classify the susceptibility/resistance of the NNRTI-mutant PSVs downstream. An unpaired Student t-test with Welch’s correction was used to compare the DOR susceptibility of NSX and MJ4: IC_50_ = *p*-value < 0.0001 (****) and FC = *p*-value > 0.9999. ns—no significance (*p*-value > 0.05), IC_50_—50% inhibitory concentration.

**Figure 2 viruses-16-01493-f002:**
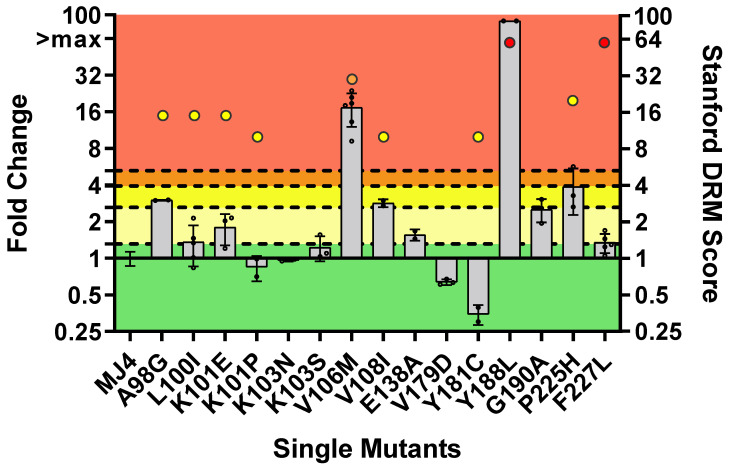
Phenotypic susceptibility of single NNRTI mutations to DOR. In vitro phenotypic assays were performed to assess the susceptibility of each mutant. The lower TCO for DOR susceptibility in the assay was 1.31. DOR susceptibility/resistance was classified as follows: FC ≤ TCO = susceptible (■); TCO > FC ≤ 2 × TCO = potential low-level resistance (■); 2 × TCO > FC ≤ 3 × TCO = low-level resistance (■); 3 × TCO > FC ≤ 4 × TCO = intermediate resistance (■); and FC > 4 × TCO = high-level resistance (■). The coloured dots above each bar graph represent Standford’s HIV Drug Resistance Mutation (DRM) score. It indicates the level of resistance predicted by their algorithm for a particular mutation (low-level resistance (■), intermediate resistance (■), high-level resistance (■)). The bars without dots (e.g., K103N, V179D) were predicted to be susceptible to DOR; as their score would be 0, the dots are not depicted. V106M and Y188L displayed high-level resistance to DOR. F227L showed low-level DOR resistance despite having a predicted high-level resistance.

**Figure 3 viruses-16-01493-f003:**
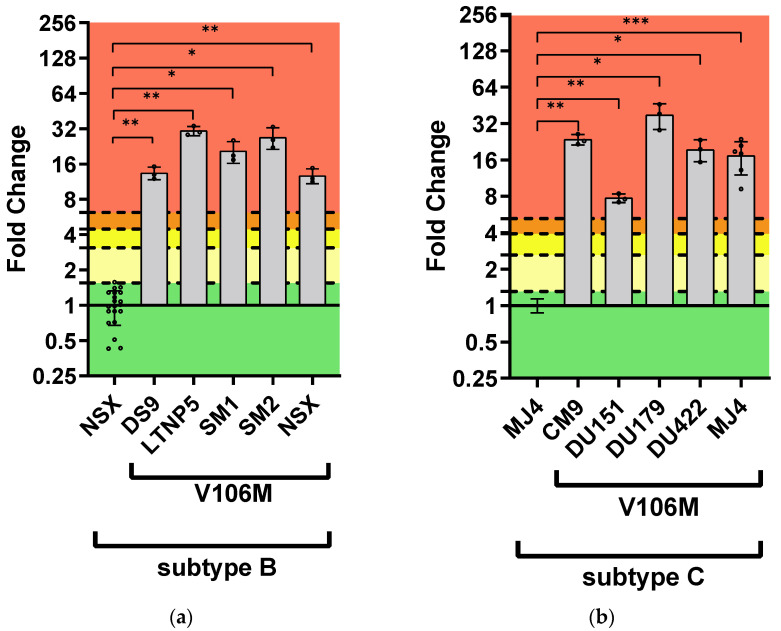
Phenotypic susceptibility of V106M in laboratory-adapted strains to DOR. In vitro phenotypic assays were performed to assess the susceptibility of V106M in the subtype B (**a**) and subtype C (**b**) laboratory-adapted strains and were expressed as FC. The lower TCO for DOR susceptibility in the assay was 1.31 for subtype C and 1.55 for subtype B. DOR susceptibility/resistance was classified as follows: FC ≤ TCO = susceptible (■); TCO > FC ≤ 2 × TCO = potential low-level resistance (■); 2 × TCO > FC ≤ 3 × TCO = low-level resistance (■); 3 × TCO > FC ≤ 4 × TCO = intermediate resistance (■); and FC > 4 × TCO = high-level resistance (■). The Stanford HIV Drug Resistance Database predicted intermediate resistance. The subtype B and C laboratory-adapted strains displayed high-level resistance to DOR. *—*p*-value ≤ 0.05, **—*p*-value ≤ 0.01, ***—*p*-value ≤ 0.001.

**Figure 4 viruses-16-01493-f004:**
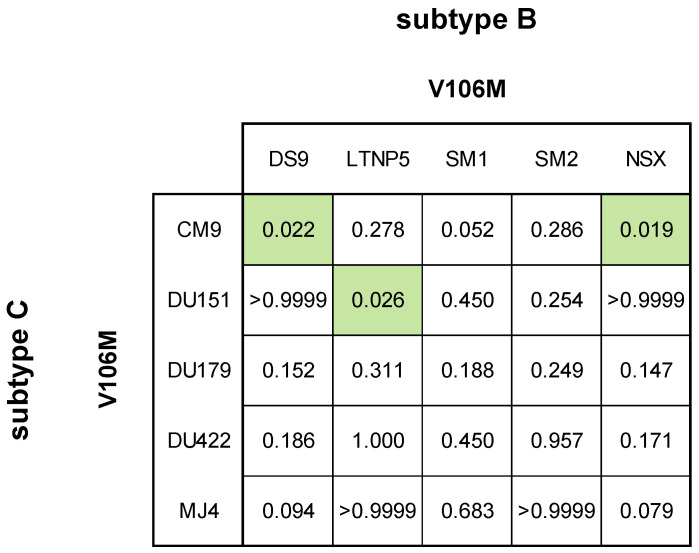
Comparing V106M susceptibility in subtype B and C laboratory-adapted strains. The phenotypic responses between the laboratory-adapted strains containing the V106M mutation were compared to each other. The mean IC_50_ of each laboratory-adapted strain was compared with a one-way ANOVA and the *p*-values < 0.05 were considered significant (■).

**Figure 5 viruses-16-01493-f005:**
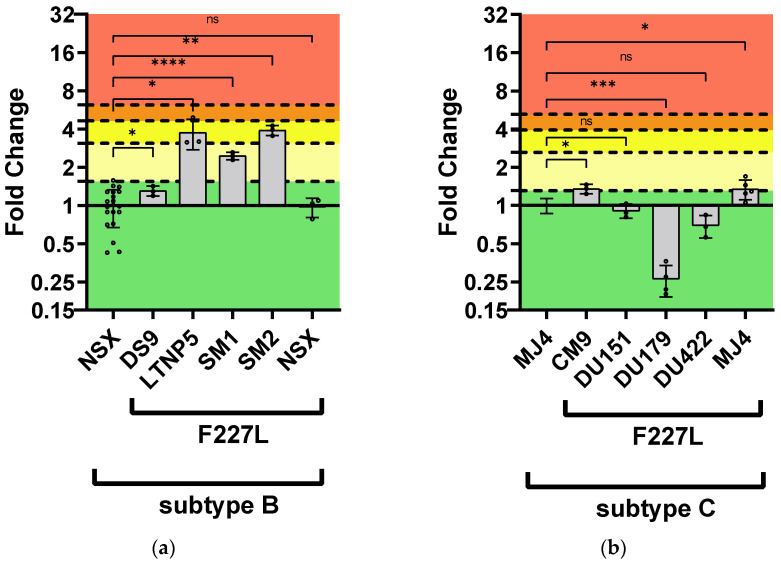
Phenotypic susceptibility of F227L in laboratory-adapted strains to DOR. In vitro phenotypic assays were performed to assess the susceptibility of F227L in the subtype B (**a**) and subtype C (**b**) laboratory-adapted strains and were expressed as FC. The lower TCO for DOR susceptibility in the assay was 1.31 for subtype C and 1.55 for subtype B. DOR susceptibility/resistance was classified as follows: FC ≤ TCO = susceptible (■); TCO > FC ≤ 2 × TCO = potential low-level resistance (■); 2 × TCO > FC ≤ 3 × TCO = low-level resistance (■); 3 × TCO > FC ≤ 4 × TCO = intermediate resistance (■); and FC > 4 × TCO = high-level resistance (■). Intermediate resistance was predicted by the Stanford HIV Drug Resistance Database. The subtype B laboratory-adapted strains displayed potential and low-level resistance to DOR. The subtype C laboratory-adapted strains displayed susceptibility to DOR. ns—no significance (*p*-value > 0.05), *—*p*-value ≤ 0.05, **—*p*-value ≤ 0.01, ***—*p*-value ≤ 0.001, ****—*p*-value ≤ 0.0001.

**Figure 6 viruses-16-01493-f006:**
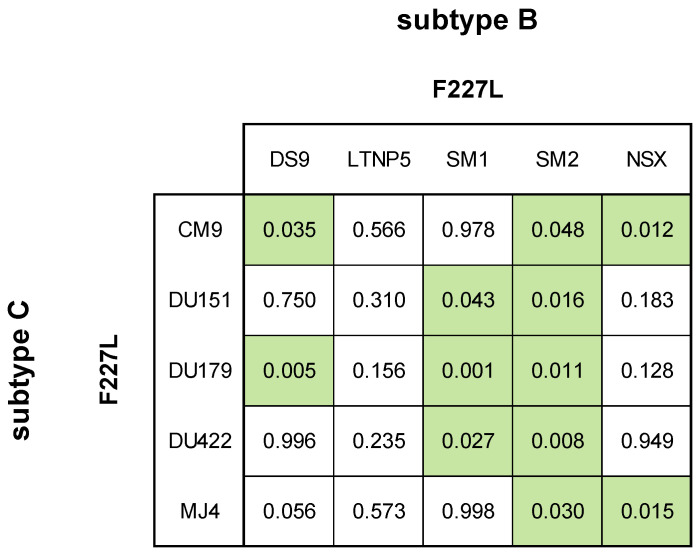
Comparing F227L susceptibility in subtype B and C laboratory-adapted strains. The phenotypic responses between the laboratory-adapted strains containing the F227L mutation were compared to each other. The mean IC_50_ of each laboratory-adapted strain was compared with a one-way ANOVA and the *p*-values < 0.05 were considered significant (■).

**Figure 7 viruses-16-01493-f007:**
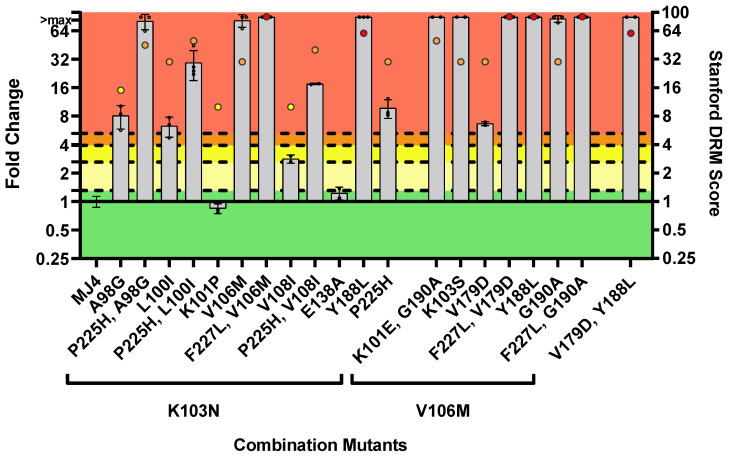
Phenotypic susceptibility of combination NNRTI mutations to DOR. In vitro phenotypic assays were performed to assess the susceptibility of each mutant and susceptibility was expressed as FC. The lower TCO for DOR susceptibility in the assay was 1.31. DOR susceptibility/resistance was classified as follows: FC ≤ TCO = susceptible (■); TCO > FC ≤ 2 × TCO = potential low-level resistance (■); 2 × TCO > FC ≤ 3 × TCO = low-level resistance (■); 3 × TCO > FC ≤ 4 × TCO = intermediate resistance (■); and FC > 4 × TCO = high-level resistance (■). The coloured dots above each bar graph represent Standford’s HIV Drug Resistance score for the Drug Resistance Mutation (DRM) profile. It indicates the level of resistance predicted by their algorithm for a particular mutation (low-level resistance (■), intermediate resistance (■), high-level resistance (■)). The majority of the mutants (17 of 20) displayed high-level resistance to DOR.

## Data Availability

The original contributions presented in the study are included in the article/Appendix A, further inquiries can be directed to the corresponding authors.

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
