# Peer review of "K103N, V106M and Y188L Significantly Reduce HIV-1 Subtype C Phenotypic Susceptibility to Doravirine"

_viruses, 2024, doi:10.3390/v16091493_

Round 1
Reviewer 1 Report
Comments and Suggestions for Authors
The manuscript: "K103N, V106M and Y188L significantly reduce HIV-1 subtype C phenotypic susceptibility to Doravirine" is overall well written and covers an important issue of monitoring the drug resistance for the new generations of antiretroviral therapy. Introduction provides sufficient information on Doravirine, but for the readers not so deep into the subject, a few more sentences on the antiretroviral therapy and resistance in general would not hurt. The Materials and methods contain detailed descriptions of how the data for this manuscript were obtained and need no corrections. The presentation of the results is clear and detailed while discussion has compared this study results to reasonable number of other data. Conclusions are supported by the results. Other than that I find the article suitable for the publishing.
Comments on the Quality of English LanguageEnglish language requires only minor intervention, resolvable within editing process, just a few changes to, for instance, avoid unnecessary repetition, like in the sentence in lines 281-282.
Reviewer 2 Report
Comments and Suggestions for Authors
The study conducted in South Africa focuses on the analysis of how NNRTI HIV drug resistance mutations affect the susceptibility of HIV-1 subtype C infected patients to doravirine (DOR). To investigate this, the most common drug resistance mutations were introduced into replication-defective pseudoviruses, and DOR susceptibility was assessed in vitro.
The study employed highly advanced methodologies, which are both complex and commendable. The authors confirmed that the single V106M and Y188L mutations led to high-level resistance, while the F227L mutation did not result in significant DOR resistance in subtype C. In contrast, strains containing the F227L mutation were found to remain susceptible to DOR in vitro.
The most critical and compelling aspect of the research, in my opinion, is the investigation of mutation combinations, including K103N, V106M, or Y188L. These mutations are frequently observed in patients infected with the subtype C virus who have experienced NNRTI treatment failure in South Africa. The study found that these combinations can substantially reduce the virus's sensitivity to DOR. The authors make a bold and important conclusion that DOR should not be prescribed to such patients without prior analysis of the HIV genotype.
This research is particularly valuable because it demonstrates, with clear evidence, the differing drug sensitivities between various HIV subtypes. Although this concept has been discussed for some time, there has been limited concrete data to support it. Given that non-B subtypes dominate in countries such as South Africa and India, where the majority of the world’s HIV-infected population resides, this information is crucial for implementing the most effective treatment strategies in those regions. Furthermore, other countries where non-B HIV subtypes are prevalent would also benefit from this kind of research.
I encourage the authors to continue their studies and expand them to include other HIV variants.
I have no critical comments on this text, as I did not find any flaws in the article. I believe it should be considered a response to the publication application rather than a standard review.
Author Response
We thank the Reviewer for taking the time to review our submitted manuscript.
Reviewer 3 Report
Comments and Suggestions for Authors
Doravirine is a second generation NNRTI, designed to be reactive against the viruses harboring K103N or Y181C mutations conferring resistance to efavirenz and nevirapine. The study by Reddy et al, assessed emerging resistance to doravirine in South Africa where subtype C is endemic subtype C. Notably, K103N and V106M combinations were frequently present in combinations (16/20 and 9/20, respectively,). The strength of this study is the design of recombinant subtype B (NSX) and subtype C (MJ4) pseudo-virus constructs assays to elaborate the differential in vitro phenotypic resistance to NNRTIs, assessing the impact of K103N, V06M and Y188l mutational combinations on in vitro resistance. emergent to doravirine in perused
The findings show that V106M and Y188L resulted in high phenotypic resistance to doravirine that was amplified in NNRTI combinations. Notably, F227L did not result in reduced susceptibility as predicted in Stanford algorithms. These findings are notable and may improve predictive algorithms for drug resistance. Although facilitated resistance to V106M is expected in subtype C is expected, the results for F227L are worth further investigation.
It would be important in future studies to analyze phenotypic resistance to other drugs such as, rilpivirine and etravirine, to better understand and compare resistance pathways to second generation NNRTIs. This should be added to the discussion. The authors should also note that although phenotypic assays allow the direct comparison of subtype B and C isolates, resistance conferred by clinical isolate may be different.
Comments on the Quality of English LanguageThere are several minor typos, e.g. F227L, line 410, that can be addressed by the text editorial team.
